# On the Importance of Backbone to the Adversarial Robustness of Object Detectors

## Abstract

Object detection is a critical component of various security-sensitive applications, such as autonomous driving and video surveillance. However, existing object detectors are vulnerable to adversarial attacks, which poses a significant challenge to their reliability and safety. Through experiments, first, we found that existing works on improving the adversarial robustness of object detectors give a false sense of security. Second, we found that using adversarially pre-trained backbone networks was essential for enhancing the adversarial robustness of object detectors. We then proposed a simple yet effective recipe for fast adversarial fine-tuning on object detectors with adversarially pre-trained backbones. Without any modifications to the structure of object detectors, our recipe achieved significantly better adversarial robustness than previous works. Finally, we explored the potential of different modern object detectors to improve adversarial robustness using our recipe and demonstrated interesting findings, which inspired us to design several state-of-the-art (SOTA) robust detectors with faster inference speed. Our empirical results set a new milestone for adversarially robust object detection. Code and trained checkpoints will be publicly available[1].

## 1 Introduction

Deep learning-based classifiers can be easily fooled by inputs with deliberately designed perturbations, *a.k.a.*, adversarial examples (Szegedy et al., 2014). To alleviate this threat, many efforts have been devoted to improving the adversarial robustness of classifiers (Athalye et al., 2018; Madry et al., 2018; Dong et al., 2020; Pang et al., 2021; Li et al., 2023a;b; Liu et al., 2023). As a more challenging task, object detection requires simultaneously classifying and localizing all objects in an image. Inevitably, object detection also suffers from adversarial examples (Xie et al., 2017; Liang et al., 2021; Zhu et al., 2021), which could lower the detection accuracy of detectors to near *zero* average precision (AP). Object detection is a fundamental task in computer vision and has plenty of security-critical real-world applications, such as autonomous driving (Arnold et al., 2019) and video surveillance (Kumar et al., 2020). Hence, it is also imperative to improve the adversarial robustness of object detectors.

In contrast to extensive studies on classifiers, improving the adversarial robustness of object detectors remains under-explored. One intuitive idea is to incorporate adversarial training (AT) (Madry et al., 2018) into object detectors. This has been done in some recent works (*e.g.*, MTD (Zhang & Wang, 2019), CWAT (Chen et al., 2021), and AARD (Dong et al., 2022)). However, by re-evaluating these works in a strong attack setting, we found that their reported adversarial robustness was overestimated with a false sense of security. For example, although claimed to be quite robust, AARD was easily evaded by our attack.

Let us recap the prevailing design principle for object detectors. Object detectors typically comprise two components: a detection-agnostic backbone network, *e.g.*, ResNet (He et al., 2016), and several detection-specific modules, *e.g.*, FPN (Lin et al., 2017a) or detection heads (Ren et al., 2015; Tian et al., 2019). Object detectors typically adopt a pre-training paradigm where the backbone network is first pre-trained on large-scale upstream classification datasets such as ImageNet (Deng et al., 2009), followed by fine-tuning the entire detector on the downstream object detection datasets. With this paradigm, object detection has benefited greatly from much data for classification. Nevertheless,

---

[1]Code is anonymously available at `https://anonymous.4open.science/r/ICLR_code-F3E4`

to improve the adversarial robustness of object detectors, existing methods (*e.g.*, MTD, CWAT, and AARD) usually used backbones benignly pre-trained (*i.e.*, pre-trained on clean examples) on upstream classification datasets and performed AT only on the downstream detection datasets. This paradigm for adversarial robustness could be sub-optimal. Firstly, the backbones pre-trained on benign examples themselves are vulnerable to adversarial examples and lack robustness (Madry et al., 2018), and thus they cannot be expected to enhance adversarial robustness on downstream tasks. Secondly, AT is data hungry and requires to be performed on a large-scale dataset (possibly exponential) to significantly improve robustness (Schmidt et al., 2018; Gowal et al., 2021; Li et al., 2022a), whereas detection datasets are usually small-scale. Different from the paradigm of existing methods, a possible alternative strategy is to use the backbone *adversarially* pre-trained on the large-scale upstream classification datasets, but to the best of our knowledge, the transferability of adversarial robustness of backbones to that of downstream tasks has never been investigated yet.

In this work, we validated the transferability and found that backbones adversarially pre-trained on the upstream dataset are essential for enhancing the adversarial robustness of object detectors. With adversarially pre-trained backbones, we proposed a new training recipe for fast adversarial fine-tuning on object detectors. Without any modifications to the structure of object detectors, our new recipe significantly surpassed previous methods on both benign accuracy and adversarial robustness, with a training cost similar to the standard training. Moreover, we investigated the potential of different modern object detectors in improving adversarial robustness with this recipe. Our empirical results revealed that *from the perspective of adversarial robustness, backbone networks play a more important role than detection-specific modules.* Inspired by this conclusion, we further designed several robust detectors with SOTA adversarial robustness and faster inference speed. Our study sets a new milestone for the adversarial robustness of detectors and highlights the need for better upstream adversarial pre-training and downstream adversarial fine-tuning techniques.

The main contributions of this work are as follows:

- We revealed the importance of adversarially pre-trained backbones, which has been long neglected by existing works, and we proposed a new recipe for fast training much more robust object detectors.

- We performed a comprehensive investigation on the structures of object detectors and we designed several SOTA robust object detectors based on the above finding.

## 2 RELATED WORKS AND PRELIMINARIES

**Object detection.** Modern object detectors consist of two main components: a detection-agnostic backbone for feature extraction and detection-specific modules (*e.g.*, necks and heads) for the detection task. The design of backbones is generally decoupled from the detection-specific modules and evolves in parallel. The detection-specific module varies depending on the detection method, which can be broadly categorized as two-stage and one-stage. Two-stage detectors regress the bounding box repeatedly based on box proposals, typically produced by RPN (Ren et al., 2015; Cai & Vasconcelos, 2021). In contrast, one-stage methods directly predict the bounding boxes with anchor boxes or anchor points, referred to as anchor-based (Lin et al., 2017b) or anchor-free (Tian et al., 2019) methods, respectively. Recently, detection transformer (DETR) (Carion et al., 2020), which models object detection as a set prediction task, has emerged as a new paradigm for object detection. To provide a comprehensive benchmark, we cover various detectors extensively.

**Adversarial robustness on classifiers.** Adversarial examples are first discovered on classifiers (Szegedy et al., 2014). Given an image-label pair $(\mathbf{x}, y)$ and a classifier $f_\theta(\cdot)$, an attacker can easily find an imperceptible adversarial perturbation $\delta$ that fools $f_\theta(\cdot)$ by maximizing the output loss: $\delta = \arg\max_{||\delta||_p \leq \epsilon} \mathcal{L}(f_\theta(\mathbf{x} + \delta), y)$, where $\mathcal{L}$ denotes the classification loss, *e.g.*, cross entroy (CE) loss, and $\epsilon$ bounds the perturbation intensity. As it is intractable to solve this maximizing problem directly, several approximate methods (Goodfellow et al., 2015; Carlini & Wagner, 2017; Madry et al., 2018) have been proposed. Among them, PGD (Madry et al., 2018) is one of the most popular attacks by iteratively taking multiple small gradient updates: $\delta_{t+1} = \text{clip}_\epsilon(\delta_t + \alpha \cdot \text{sign}(\nabla_{\delta_t} \mathcal{L}))$, where $\alpha$ denotes the step size. Adversarial training and its variants are generally recognized as the most effective defense methods against adversarial examples, which improve the adversarial robustness of

classifiers by incorporating adversarial examples into training:

$$\theta = \arg\min_\theta \mathbb{E}_{\mathbf{x}}\{\max_{||\delta||_p \leq \epsilon} \mathcal{L}(f_\theta(\mathbf{x} + \delta), y)\}. \tag{1}$$

**Adversarial robustness on object detectors.** Object detectors are also fragile to adversarial examples and many attacks on detectors have been proposed (Xie et al., 2017; Zhang & Wang, 2019; Liang et al., 2021; Zhu et al., 2021; Chen et al., 2021). To improve the safety of object detectors, one intuitive idea is to adjust the AT strategy on classifiers to object detection tasks. This can be achieved by replacing the classification loss $\mathcal{L}$ in Eq. (1) with the detection loss $\mathcal{L}_d$. Given an image $\mathbf{x}$ with $K$ bounding box labels $\{y_i, \mathbf{b}_i\}_{i=1}^K$, the loss $\mathcal{L}_d$ is:

$$\mathcal{L}_d = \mathcal{L}_{\mathrm{cls}} + \mathcal{L}_{\mathrm{reg}} = \sum_{i=1}^K l_{\mathrm{cls}}(\hat{y}_i, y_i) + \sum_{i=1}^K l_{\mathrm{reg}}(\hat{\mathbf{b}}_i, \mathbf{b}_i), \tag{2}$$

where $\hat{y}_i$ and $\hat{\mathbf{b}}_i$ denote the output of detectors, $l_{\mathrm{cls}}$ can be a CE loss for classification and $l_{\mathrm{reg}}$ can be a $L_1$ loss for regression. As $\mathcal{L}_d$ consists of multiple terms, the generation of adversarial examples can take various forms, *e.g.*, maximizing $\mathcal{L}_{\mathrm{cls}}$ only. To find adversarial examples more suitable for AT, MTD (Zhang & Wang, 2019) formulates it to be a multi-task problem and maximizes $\mathcal{L}_{\mathrm{mtd}} = \sum_{i=1}^K \{\max\{l_{\mathrm{cls}}(\hat{y}_i, y_i), l_{\mathrm{reg}}(\hat{\mathbf{b}}_i, \mathbf{b}_i)\}\}$ to generate adversarial examples for AT. CWAT (Chen et al., 2021) improves vanilla loss for AT ($\mathcal{L}_d$) by generating examples with the class-wise attack (CWA), which takes the class imbalance problem of object detection into account and maximizes $\mathcal{L}_{\mathrm{cwa}} = \sum_{i=1}^K w_i \cdot l_{\mathrm{cls}}(\hat{y}_i, y_i) + \sum_{i=1}^K w_i \cdot l_{\mathrm{reg}}(\hat{\mathbf{b}}_i, \mathbf{b}_i)$, where $w_i$ denotes a weight with respect to the number of each class in an image. Recently, AARD (Dong et al., 2022) uses an adversarial image discriminator to distinguish benign and adversarial images and optimizes different parts of the network with AT and standard training together. However, all these works did not adversarially pre-train the backbones. Besides these empirical methods, Chiang et al. (2020) investigates certified defense for object detectors, but till now the certified methods only work with quite tiny perturbations.

# 3 RE-EVALUATION ON PREVIOUS METHODS

In this section, we describe our evaluation method. With a strong attack setting, we re-evaluated the adversarial robustness of models trained in previous studies (Zhang & Wang, 2019; Chen et al., 2021; Dong et al., 2022).

## 3.1 ATTACK SETTINGS

All attacks were considered under the most commonly used norm-ball $||\mathbf{x} - \mathbf{x}_{\mathrm{adv}}||_\infty \leq \epsilon/255$, which bounded the maximal difference for each pixel of an image $\mathbf{x}$. PGD with 20 iterative steps in the white-box setting was performed under the attack intensity.

We note that previous works evaluated their methods only in a mild attack setting, considering only FGSM (Goodfellow et al., 2015) and PGD (Madry et al., 2018) attacks with a step size $\alpha$ equal to the intensity $\epsilon$. Instead, following the *AutoAttack* (AA) benchmark (Croce & Hein, 2020b) on reliable evaluation of image classifiers, we chose the PGD step size $\alpha$ as $\epsilon/4$, which achieved the best attack performance among different step sizes $\epsilon/10, \epsilon/4, \epsilon/2, \epsilon$. We did not use AA directly as its inference speed on object detectors is quite slow and some of its attacks (*e.g.*, FAB (Croce & Hein, 2020a)) are designed specifically for classification. As discussed in Sec. 2, adversarial examples for object detectors can be generated by maximizing different losses. Thus following previous studies, we evaluated the robustness with the three attacks: 1) $A_{\mathrm{cls}}$: Maximizing the classification loss $\mathcal{L}_{\mathrm{cls}}$ only (Zhang & Wang, 2019); 2) $A_{\mathrm{reg}}$: Maximizing the regression loss $\mathcal{L}_{\mathrm{reg}}$ only (Zhang & Wang, 2019); 3) $A_{\mathrm{cwa}}$: Maximizing the classification and regression losses simultaneously with class imbalance problem (Chen et al., 2021) considered (*i.e.*, maximizing $\mathcal{L}_{\mathrm{cwa}}$). Note that these attacks were all implemented using PGD with 20 steps and $\alpha = \epsilon/4$.

## 3.2 RE-EVALUATION RESULTS

Following the main setting of previous works (Zhang & Wang, 2019; Chen et al., 2021; Dong et al., 2022), we used the PASCAL VOC (Everingham et al., 2015) dataset for re-evaluation. The standard

Table 1: The evaluation results of several methods with the original training recipe (the benignly pre-trained backbone) and our training recipe under various adversarial attacks on PASCAL VOC.

| Method | SSD | | | | Faster R-CNN | | | |
|---|---|---|---|---|---|---|---|---|
| | Benign | $A_{\mathrm{cls}}$ | $A_{\mathrm{reg}}$ | $A_{\mathrm{cwa}}$ | Benign | $A_{\mathrm{cls}}$ | $A_{\mathrm{reg}}$ | $A_{\mathrm{cwa}}$ |
| STD | 76.2 | 1.3 | 5.3 | 1.4 | 80.4 | 0.1 | 0.2 | 0.0 |
| MTD (Zhang & Wang, 2019) | 55.3 | 19.6 | 38.1 | 19.6 | 60.0 | 18.2 | 39.7 | 20.7 |
| CWAT (Chen et al., 2021) | 54.2 | 21.0 | 38.5 | 20.4 | 58.2 | 19.1 | 39.8 | 20.8 |
| AARD (Dong et al., 2022) | 75.4 | 0.7 | 3.9 | 1.0 | - | - | - | - |
| VANAT | 54.8 | 20.7 | 37.7 | 20.3 | 58.5 | 19.0 | 40.3 | 21.8 |
| MTD w/ Our Recipe | 58.3 | 25.1 | 44.5 | 25.1 | 70.0 | 30.8 | 51.4 | 33.2 |
| CWAT w/ Our Recipe | 57.4 | **27.7** | **44.9** | **26.1** | 69.0 | **32.2** | 51.7 | 33.7 |
| VANAT w/ Our Recipe | 58.2 | 25.2 | 44.8 | 24.7 | 69.7 | **32.2** | **51.8** | **34.4** |

"07+12" protocol was adopted for training, containing 16,551 images of 20 categories. The PASCAL VOC 2007 `test` set was used during testing, which includes 4,952 test images. We report the PASCAL-style $\mathrm{AP}_{50}$, which was computed at a single Intersection-over-Union (IoU) threshold of 0.5. The attack intensity was set to be $\epsilon = 8$ here.

Previous works only evaluated their methods on the early object detector SSD (Liu et al., 2016) at a relatively low input resolution. In this study, we replicated the methods of MTD and CWAT using the Faster R-CNN (Ren et al., 2015) at a higher input resolution. The Faster R-CNN was implemented with FPN (Lin et al., 2017a) and ResNet-50 (He et al., 2016). Each object detector was first pre-trained on the benign images of PASCAL VOC, denoted as standard method (*STD*), and then AT was performed using the methods of MTD, CWAT, and AARD on the pre-trained STD models. We also performed AT with the adversarial examples generated by attacking the original $L_d$ (see Sec. 2) for comparison, denoted as *VANAT* (vanilla loss of detectors for AT). Following the original settings, SSD was adversarially trained for 240 epochs and Faster R-CNN was adversarially trained for 24 epochs (*i.e.*, $2\times$ schedule). More implementation details are shown in Appendix A.

The first five rows of Table 1 show the evaluation results of these methods in the unified attack settings. Obviously, the STD detectors were highly vulnerable to adversarial attacks, with their $\mathrm{AP}_{50}$ reduced to nearly zero. CWAT and MTD did not show significant improvements over VANAT under the attack with the small step size. And regretfully, although AARD claimed 41.5% $\mathrm{AP}_{50}$ under $A_{\mathrm{cls}}$ in the original paper, it showed even worse robustness against these attacks than STD. Note that the attacks were entirely based on their released code and checkpoints[2] under the same attack intensity $\epsilon = 8$, with only the PGD step size $\alpha$ changed. By scrutinizing the AARD approach, we found that their adversarial discriminator worked only with large perturbation magnitudes, yet several small perturbation updates could easily bypass it.

## 4 THE IMPORTANCE OF ADVERSARIALLY PRE-TRAINED BACKBONES

We first introduce a new training recipe for fast AT on object detectors, then demonstrate the importance of adversarially pre-trained backbones for object detection with this recipe. Finally, we describe ablation studies to analyze the effectiveness of each component of the recipe.

### 4.1 A NEW TRAINING RECIPE

Previous works (Zhang & Wang, 2019; Chen et al., 2021; Dong et al., 2022) neglected the importance of adversarially pre-trained backbones and used benignly pre-trained backbones. Here we propose a new training recipe for building adversarially robust object detectors based on the upstream adversarially pre-trained backbones. The customized recipe is summarized as follows:

1. Initialize the object detector with backbones *adversarially* pre-trained on the upstream classification dataset;
2. Fine-tune the whole detector with *adversarial training* on the downstream object detection dataset using an *AdamW optimizer* with a *smaller learning rate* for the backbone network.

---

[2] `https://github.com/7eu7d7/RobustDet`

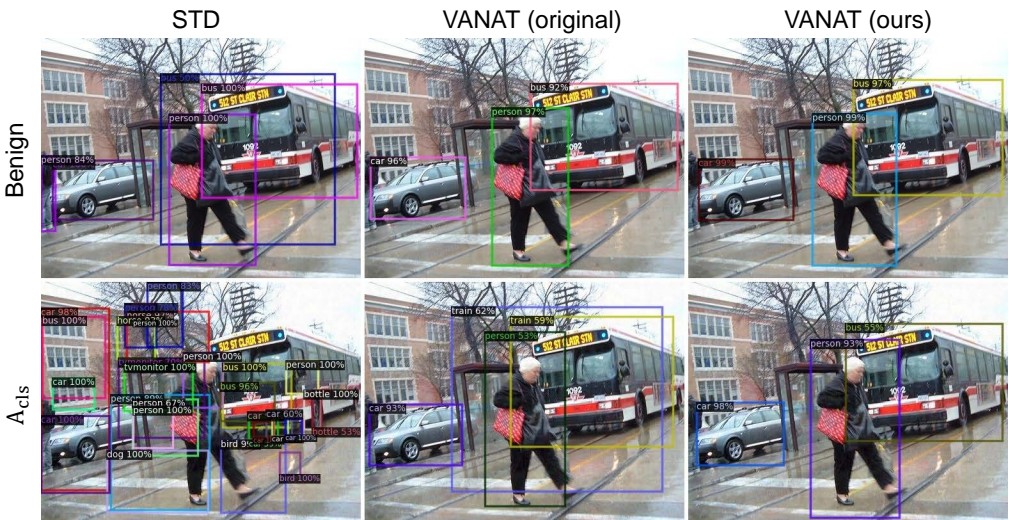

Figure 1: Visualization of the detection results on benign images (upper) and $A_{cls}$ adversarial images (lower), with three training methods STD (left), VANAT with the recipe of previous work (medium), and VANAT with our recipe (right). Faster R-CNN models were used here.

The other settings *default to* the standard setups of the corresponding detectors. Our intention here is to follow the basic training paradigm of detectors and keep the recipe as concise as possible so that it can be more scalable and generalizable. We did not use any customized method like continual learning techniques (De Lange et al., 2021). Any modifications to the structure of object detectors were not performed, either. We present each component of this recipe in turn.

**Upstream Adversarial Pre-training.** On benign images, object detection has benefited greatly from backbones benignly pre-trained on large upstream datasets. We argue that the adversarial robustness of object detection can also benefit greatly from backbones adversarially pre-trained on large upstream datasets. Considering that quite a lot of models adversarially pre-trained on upstream datasets such as ImageNet are publicly available (Salman et al., 2020; Tang et al., 2021; Debenedetti et al., 2022; Liu et al., 2023), with our recipe, employing them to improve the robustness of object detectors is almost free. The cost of adversarial pre-training is further discussed in Appendix B.

**Downstream Adversarial Fine-tuning.** Due to the high computational cost of AT, we opted for FreeAT (Shafahi et al., 2019) as the default AT method for object detection. Unlike the full PGD-AT (Madry et al., 2018), which requires multiple iterative steps for one gradient update, FreeAT recycles gradient perturbations to reduce extra training costs brought by AT while achieving comparable adversarial robustness. We set the batch replay parameter $m$ for FreeAT to 4. The pseudo-code of FreeAT on object detection is provided in Appendix C.

**Learning Rate and Optimizer.** To ensure that the original adversarial robustness of backbones is preserved during downstream fine-tuning and the detection-specific modules can be trained in the usual way, we decay the learning rate of the backbone by a factor of 0.1 when performing AT on object detectors. In addition, although many recent works (Pang et al., 2021; Mo et al., 2023) suggest that using SGD optimizer with momentum in AT can obtain better adversarial robustness for classifiers, we used the AdamW (Loshchilov & Hutter, 2019) optimizer. This is motivated by the fact that modern detectors, *e.g.*, DETR, tend to use AdamW to achieve better detection accuracy.

## 4.2 RESULTS WITH THE NEW RECIPE

We used our recipe to adversarially train several detectors by MTD, CWAT, and VANAT. The adversarially pre-trained ResNet-50 from Salman et al. (2020) was used as the backbone here. Unless otherwise specified, other settings were the same as described in Sec. 3. The evaluation results of these models are shown in the last three rows of Table 1. Our recipe significantly outperformed previous methods on both benign examples and different adversarial examples. For SSD, our recipe achieved 27.2% $AP_{50}$ under $A_{cls}$ with CWAT, resulting in a **6.7%** $AP_{50}$ improvement. For Faster

Table 2: The evaluation results of Faster R-CNN trained with different recipes on PASCAL VOC.

| Pre-training Method | | | Optimizer | | Backbone | Schedule | Benign | $A_{cls}$ | $A_{reg}$ | $A_{cwa}$ |
| U-Beni. | D-Beni. | U-Adv. | SGD | AdamW | LR | | | | | |
|---|---|---|---|---|---|---|---|---|---|---|
| ✓ | | | ✓ | | | | 44.6 | 15.7 | 34.6 | 16.4 |
| ✓ | | | | ✓ | 1× | 2× | 48.4 | 18.3 | 36.2 | 20.0 |
| | ✓ | | ✓ | | | | 58.5 | 19.0 | 40.3 | 21.8 |
| | ✓ | | | ✓ | | | 54.2 | 20.0 | 39.1 | 22.2 |
| | | ✓ | ✓ | | 1× | | 64.7 | 29.0 | 49.0 | 31.8 |
| | | ✓ | ✓ | | 0× | | 61.9 | 28.8 | 47.6 | 31.2 |
| | | ✓ | | ✓ | 1× | 2× | 54.2 | 21.2 | 40.4 | 23.8 |
| | | ✓ | | ✓ | 0× | | 64.5 | 30.0 | 49.9 | 32.1 |
| | | ✓ | ✓ | | 0.1× | | 67.9 | 31.1 | 51.5 | 33.6 |
| | | ✓ | | ✓ | 0.1× | | 69.7 | **32.2** | **51.8** | **34.4** |
| | | ✓ | | ✓ | 0.1× | 4× | **70.1** | 31.2 | 50.8 | 33.2 |

(a) Benign $AP_{50}$.     (b) $AP_{50}$ under $A_{cls}$.

Figure 2: Evaluation results of detectors in various epoch settings on the PASCAL VOC. (a) $AP_{50}$ on benign images. (b) $AP_{50}$ under $A_{cls}$. Here the models were initialized by downstream benignly pre-trained backbones except for the red dashed line, which denotes the performance of the model trained by our overall recipe (24 epochs).

R-CNN, the gains were even above **10**% $AP_{50}$ due to the higher input resolution. The visualization comparisons in Fig. 1 and Appendix D.1 show that the model with our recipe performed significantly better with more objects correctly detected under attack.

### 4.3 ABLATION STUDY

We conducted ablation experiments on Faster R-CNN with VANAT to verify the effectiveness of our training recipe. We compared three pre-training methods: upstream benign pre-training, downstream benign pre-training (using pre-trained STD models), and upstream adversarial pre-training, denoted as *U-Beni.*, *D-Beni.* and *U-Adv.*, respectively. Three learning rate settings for the backbone networks were also compared: using the standard learning rate of object detectors (1×), using 0.1× standard learning rate, and freezing the whole backbone network (0×). The results are shown in Table 2. Clearly, upstream adversarial pre-training is vital to the adversarial robustness of object detectors, and other settings like the backbone learning rate scaling in our recipe are also important. The last row of Table 2 shows that further extending the training schedule brought modest gains.

In addition, as shown in Fig. 2, training longer with the benignly pre-trained backbone models slightly improved adversarial robustness. However, the best performance is still far from our recipe with upstream adversarial pre-training. The results presented in Appendix D.2 indicate that detectors trained with our recipe for 2× achieve comparable adversarial robustness to those trained with full PGD-AT, which requires 20× training time.

## 5 INVESTIGATING ADVERSARIAL ROBUSTNESS OF MODERN DETECTORS

Previous works (Zhang & Wang, 2019; Chen et al., 2021; Dong et al., 2022) have only examined their methods on early simple detectors such as SSD (Liu et al., 2016). However, the field of object detection is rapidly developing, with many new detectors being proposed. The potential of different modern detectors to improve adversarial robustness is still unknown. Therefore, we investigated their potential with our new training recipe. Our investigation focused on detection-specific modules and detection-agnostic backbone networks. Since object detection has benefited from many independent explorations of these two components, such investigation could also help to build more robust object detectors from the two aspects.

Table 3: The evaluation results of object detectors under VANAT (two different training recipes, Beni-AT and Our-AT) and standard training (STD) on MS-COCO. The results of $AP_{50}$ are shaded as it is a more practical metric. More results of $A_{reg}$ and $A_{cwa}$ are shown in Appendix E.2.

| Detector | Method | Benign | | | | | | $A_{cls}$ | | | | | | $A_{reg}$ | $A_{cwa}$ |
|---|---|---|---|---|---|---|---|---|---|---|---|---|---|---|---|
| | | AP | $AP_{50}$ | $AP_{75}$ | $AP_S$ | $AP_M$ | $AP_L$ | AP | $AP_{50}$ | $AP_{75}$ | $AP_S$ | $AP_M$ | $AP_L$ | $AP_{50}$ | $AP_{50}$ |
| Faster R-CNN | STD | 40.5 | 62.2 | 44.0 | 24.3 | 44.1 | 52.6 | 0.0 | 0.1 | 0.0 | 0.0 | 0.0 | 0.1 | 0.1 | 0.0 |
| | Beni-AT | 24.4 | 41.2 | 25.5 | 13.1 | 26.3 | 31.9 | 10.6 | 18.6 | 10.7 | 4.1 | 10.7 | 15.5 | 33.7 | 22.1 |
| | Our-AT | 29.9 | 49.3 | 31.6 | 15.0 | 32.4 | 40.7 | 14.8 | **25.5** | 15.1 | 5.6 | 14.9 | 22.2 | **40.5** | **29.3** |
| FCOS | STD | 41.9 | 60.9 | 45.4 | 26.4 | 45.5 | 54.4 | 0.5 | 1.4 | 0.2 | 0.1 | 0.5 | 1.1 | 4.8 | 1.4 |
| | Beni-AT | 22.6 | 35.6 | 23.7 | 12.5 | 24.3 | 29.5 | 10.7 | 17.7 | 10.8 | 4.8 | 11.0 | 15.2 | 33.9 | 16.6 |
| | Our-AT | 30.5 | 46.6 | 32.4 | 16.4 | 33.2 | 40.8 | 15.5 | **25.2** | 15.9 | 6.4 | 16.0 | 22.4 | **44.4** | **24.0** |
| DN-DETR | STD | 41.4 | 61.9 | 43.9 | 19.4 | 45.6 | 62.0 | 0.1 | 0.2 | 0.0 | 0.0 | 0.1 | 0.2 | 6.4 | 0.5 |
| | Beni-AT | 28.4 | 44.8 | 29.9 | 10.7 | 31.4 | 44.7 | 11.0 | 18.4 | 10.7 | 3.9 | 11.5 | 17.1 | 43.6 | 17.5 |
| | Our-AT | 31.8 | 49.1 | 33.4 | 12.5 | 34.1 | 49.6 | 16.8 | **27.7** | 17.1 | 5.3 | 17.7 | 26.7 | **43.8** | **27.4** |

## 5.1 Experimental settings

The investigation was performed on the challenging MS-COCO dataset. We used the 2017 version, which contains 118,287 images of 80 categories for training and 5,000 images for the test, and reported the COCO-style AP (Lin et al., 2014) (averaged over 10 IoU thresholds ranging from 0.5 to 0.95), as well as $AP_{50}$, $AP_{75}$, and $AP_S$/$AP_M$/$AP_L$ (for small/medium/large objects). But we focused on $AP_{50}$ as it is a more practical metric for object detection (Redmon & Farhadi, 2018). Following the common attack setting on ImageNet, $\epsilon = 4$ was used.

The implementation was based on the popular MMDetection toolbox (Chen et al., 2019). Unless otherwise specified, the detectors were adversarially trained with our recipe (upstream adversarially pre-trained backbones) by $2\times$ training schedule. Training settings across the detectors are generally consistent to ensure comparability and are provided in Appendix E.1. For comparison, we also trained detectors with benignly pre-trained backbones by VANAT, denoted as *Beni-AT* (recipe of previous works). As shown in Table 3, VANAT with our recipe, denoted as *Our-AT*, achieved significantly better results than Beni-AT across various object detectors, *e.g.*, **7.5%** $AP_{50}$ gain under $A_{cls}$ on FCOS (see Sec. 5.2 for the introduction to different detectors). This conclusion is consistent with that of Table 1: *the adversarially pre-trained backbones lead to significantly robust detectors*.

Table 4: The heterogeneous characteristics of three types of object detectors.

| Detector | NMS | | Anchor | | Feature | |
|---|---|---|---|---|---|---|
| | Need | No-Need | Anchor-Based | Anchor-Free | Single-Scale | Multi-Scale |
| Faster R-CNN | ✓ | | ✓ | | | ✓ |
| FCOS | ✓ | | | ✓ | | ✓ |
| DN-DETR | | ✓ | | ✓ | ✓ | |

## 5.2 Different detection-specific modules

We then study the impact of different detection-specific modules on the robustness of object detectors. To provide a benchmark of existing detectors, we covered various methods as comprehensively as possible. Specifically, we selected three representative methods, including Faster R-CNN (Ren et al., 2015), FCOS (Tian et al., 2019), and DN-DETR (Li et al., 2022b), which respectively represent two-stage, one-stage, and DETR-like detectors. Table 4 provides a comparison of these detectors. One-stage object detectors can be classified as anchor-based or anchor-free, of which we chose the anchor-free detector (*i.e.*, FCOS) for its modernity and concision. For DETR, we selected DN-DETR for its fast convergence. Note that we followed the original DN-DETR and used single-scale features. We used ResNet-50 (He et al., 2016) as the backbone for all detectors here. The performances of these detectors are shown in Table 3. Despite the heterogeneous detection-specific modules, the detectors with upstream adversarially pre-trained backbones achieved similar detection accuracy (*i.e.*, $AP_{50}$) under attack. The results suggest that *detection-specific modules may not be a critical factor affecting the robustness when adversarially pre-trained backbones are utilized*.

In addition to the above conclusion, we also made other interesting findings with these results. We observed from Table 3 that for objects of different scales, the accuracy before and after attacks follows

a similar trend. As an example, on benign images, DN-DETR has significantly higher accuracy on large objects ($AP_L$) than others (probably due to the single-scale features), and this property was preserved after attacks. Thus we conclude that *adversarial robustness of detectors on objects with different scales depends on its corresponding accuracy on benign examples.* With strong attacks such as $A_{cls}$, all three detectors yielded poor results (*i.e.*, 5-7% AP) on small objects. This could be attributed to the fact that, as small objects are hard to detect, *the small-object-friendly designs (e.g., multi-scale features in detection-specific modules) fail to work properly under the attack.* We further analyzed the errors in Appendix E.3 and found that *the attacks mainly caused false negative (FN) errors and background errors (BG) of detectors.*

Table 5: The evaluation results of object detectors with two backbones ResNet-50 (R-50) and ConvNeXt-T (X-T) on MS-COCO. Detectors are trained by VANAT with our recipe.

| Detector | Backbone | Benign | | | | | | $A_{cls}$ | | | | | | $A_{reg}$ | $A_{cwa}$ |
|---|---|---|---|---|---|---|---|---|---|---|---|---|---|---|---|
| | | AP | $AP_{50}$ | $AP_{75}$ | $AP_S$ | $AP_M$ | $AP_L$ | AP | $AP_{50}$ | $AP_{75}$ | $AP_S$ | $AP_M$ | $AP_L$ | $AP_{50}$ | $AP_{50}$ |
| Faster R-CNN | R-50 | 29.9 | 49.3 | 31.6 | 15.0 | 32.4 | 40.7 | 14.8 | 25.5 | 15.1 | 5.6 | 14.9 | 22.2 | 40.5 | 29.3 |
| | X-T | 34.3 | 55.4 | 36.6 | 19.3 | 36.9 | 46.8 | 19.0 | **32.4** | 19.3 | 7.4 | 19.5 | 28.7 | **46.4** | **35.9** |
| FCOS | R-50 | 30.5 | 46.6 | 32.4 | 16.4 | 33.2 | 40.8 | 15.5 | 25.2 | 15.9 | 6.4 | 16.0 | 22.4 | 44.4 | 24.0 |
| | X-T | 35.6 | 53.8 | 37.7 | 20.1 | 38.2 | 48.1 | 19.8 | **31.7** | 20.5 | 8.6 | 20.2 | 29.0 | **50.8** | **30.4** |
| DN-DETR | R-50 | 31.8 | 49.1 | 33.4 | 12.5 | 34.1 | 49.6 | 16.8 | 27.7 | 17.1 | 5.3 | 17.7 | 26.7 | 43.8 | 27.4 |
| | X-T | 34.2 | 52.0 | 36.1 | 13.4 | 36.6 | 54.7 | 19.9 | **32.0** | 20.3 | 7.1 | 20.9 | 32.8 | **47.4** | **30.9** |

## 5.3 DIFFERENT BACKBONE NETWORKS

We have shown that different detection-specific modules may not be a critical factor affecting the robustness when adversarially pre-trained backbones are utilized. Now we explore the impact of different backbone networks.

First, we investigated the influence of using backbones with different upstream adversarial robustness on the adversarial robustness of detectors. We trained different detectors with two backbone networks: ResNet-50 and ConvNeXt-T (Liu et al., 2022). With a similar number of parameters as ResNet-50, ConvNeXt-T achieved better adversarial accuracy on the upstream ImageNet dataset (48.8% v.s. 36.4% under AA), due to its modern architectures (*e.g.*, enlarged kernel size and reduced activation). The evaluation results are shown in Table 5. We found that the backbone network has a significant impact on robustness, *e.g.*, for Faster R-CNN, using ConvNeXt-T has a 6.9% AP gain over using ResNet-50 under $A_{cls}$. We also investigated the influence of different upstream adversarial pre-training manners for the same backbone. The results shown in Appendix E.4 indicate that detection performance can be improved in a better adversarial pre-training manner. Taken together, we conclude that *better upstream adversarially pre-trained backbones significantly help to build more robust object detectors.*

Second, we investigated the transferability of adversarial examples over different detectors by changing backbone networks or detection-specific modules by the transfer attacks. The results are shown in Fig. 3. The left three columns of the left sub-figure have lower values than the right three columns, and the right three columns of the right sub-figure have lower values than the left three

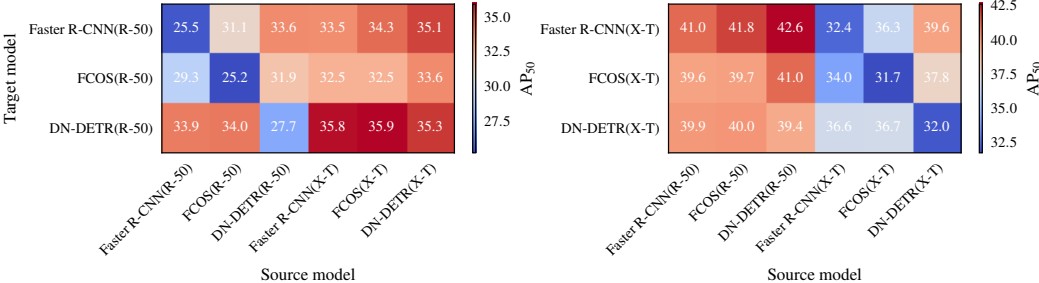

Figure 3: Black-box transferability across object detectors trained by VANAT. The adversarial examples generated on the *source* models (each column) were fed into the *target* models (each row). The values denote the $AP_{50}$ of the target models on these adversarial images. The figure is divided into two parts according to the backbone of the target model for the convenience of comparison.

columns. For a specific target model FCOS(R-50) (the 1st row of Fig. 3, left), adversarial examples from models with the same backbone network (*i.e.*, Faster R-CNN(R-50) and DN-DETR(R-50)) caused lower $AP_{50}$. Thus, we conclude that *transferring between different detection-specific modules is easier than transferring between different backbone networks*. Note that here the detection-specific modules and detection-agnostic backbones have comparable parameters, *e.g.*, DN-DETR(R-50) has about 23M/20M parameters for backbone/detection-specific modules.

## 6 APPLICATION OF THE FINDINGS

Taken together, we revealed that *from the perspective of adversarial robustness, backbone networks play a more important role than detection-specific modules*. Note that the conclusion is quite different from that on benign accuracy, where both backbones and detection-specific modules are important to improve benign accuracy (Tian et al., 2019; Li et al., 2022b). We further explore how this conclusion could be applied to build more robust models.

**Designing better robust object detectors.** Inspired by the conclusion that backbone networks play a more important role than detection-specific modules, we redesigned several object detectors towards SOTA adversarial robustness. Our design principle is to *allocate more computation to the backbone and reduce the computation of detection-specific modules so that the overall inference speed is not sacrificed* (see Appendix F for detailed configurations). As shown in Table 6, our redesigned object detectors achieved consistent improvements on benign examples and different adversarial examples among all three detectors with a faster actual inference speed (frame per second, FPS).

Table 6: Results of the redesigned detectors with ConvNeXt backbone. Symbol $*$ denotes our designed detectors with new computation allocation. Numbers in brackets show the improvement relative to the original detectors in Table 5. See Appendix F for details on computational costs.

| Detector | Benign | | $A_{cls}$ | | $A_{reg}$ | $A_{cwa}$ | FPS |
|---|---|---|---|---|---|---|---|
| | AP | $AP_{50}$ | AP | $AP_{50}$ | $AP_{50}$ | $AP_{50}$ | |
| Faster R-CNN$^*$ | 35.1(0.8) | 56.4(1.0) | 19.7(0.7) | 33.3(0.9) | 47.3(0.9) | 37.1(1.2) | 25.6(0.2) |
| FCOS$^*$ | 36.6(1.0) | 55.0(1.2) | 21.0(1.2) | 33.3(1.6) | 52.2(1.4) | 31.9(1.5) | 25.3(0.8) |
| DN-DETR$^*$ | 34.7(0.5) | 53.0(1.0) | 20.3(0.4) | 32.8(0.8) | 47.8(0.4) | 31.7(1.2) | 20.1(0.1) |

**Generalization to other tasks.** Besides object detection, the adversarial robustness of other dense prediction tasks such as image segmentation could also benefit from our conclusion. As a preliminary validation, on MS-COCO, we report the results (Table 7) on the challenging panoptic segmentation task, which requires solving both instance and semantic segmentation tasks. With our recipe, Panoptic Quality (PQ) increased significantly compared with the previous SOTA method (Daza et al., 2022). Note that our method was evaluated under $A_{cls}$, a stronger attack, and thus the gains may have been underestimated, as discussed in Appendix G.

Table 7: Results of adversarially trained segmentation models under attack. The results of Daza et al. (2022) are copied from their original paper while ours was evaluated under $A_{cls}$, a stronger attack.

| Detector | PQ | SQ | RQ |
|---|---|---|---|
| PanopticFPN (Daza et al., 2022) | 15.9 | 72.0 | 20.0 |
| PanopticFPN (Our-AT) | **20.6** | **72.6** | **26.1** |

## 7 CONCLUSION AND DISCUSSION

In this work, we highlight the importance of adversarially pre-trained backbones in achieving better adversarial robustness of object detectors. Our new training recipe with the adversarially pre-trained backbones significantly outperformed previous methods. By analyzing several heterogeneous detectors, we revealed useful and interesting findings on object detectors, which inspired us to design several object detectors with SOTA adversarial robustness. Our work establishes a new milestone in the adversarial robustness of object detection and encourages the community to explore the potential of large-scale pre-training on adversarial robustness more. As further discussed in Appendix H, we believe this study serves as a basis for building better adversarially robust object detectors in the future. The potential ethical influence is shown in Appendix I.

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

## A  OTHER IMPLEMENTATION DETAILS ON PASCAL VOC

On PASCAL VOC, SSD was trained with an input resolution of $300 \times 300$, and Faster R-CNN was trained with a higher input resolution of $1000 \times 600$. When optimized by SGD, the detectors used an initial learning rate of $1 \times 10^{-2}$ with a momentum of 0.9. When optimized by AdamW (see Sec. 4), the detectors used an initial learning rate of $1 \times 10^{-4}$. A weight decay of $1 \times 10^{-4}$ was used for all detectors on PASCAL VOC. For the learning rate schedule, SSD used multi-step decay that scaled the learning rate by 0.1 after the 192nd and 224th epochs, and Faster R-CNN used multi-step decay that scaled the learning rate by 0.1 after the 16th and 20th epochs.

## B  ABOUT THE COST OF UPSTREAM ADVERSARIAL PRE-TRAINING

Using models (benignly) pre-trained on upstream classification datasets such as ImageNet is the de facto practice for object detection together with many other downstream dense-prediction tasks. Instead, our recipe requires adversarial pre-training on upstream classification datasets. Currently, most adversarial training on ImageNet uses PGD with two (Debenedetti et al., 2022) or three (Salman et al., 2020; Liu et al., 2023) iterations. Thus the training cost of adversarial pre-training is about three or four times longer than that of benign pre-training. We believe that some fast AT methods (Shafahi et al., 2019; Wong et al., 2020) could also be used for adversarial pre-training, and then the cost for adversarial pre-training could be reduced to the same as the benign pre-training.

In addition, we found that without upstream adversarial pre-training, only extending the AT time for $10\times$ on the object detection task resulted in saturation of adversarial robustness (as discussed in Sec. 4.3), which performed significantly poorer than those trained for $2\times$ with upstream adversarial pre-training. Thus our improvements did not come from longer training time than previous works.

## C  PSEUDO-CODE OF FREEAT ON OBJECT DETECTION

The pseudo-code of FreeAT (Shafahi et al., 2019) on the object detection task is presented in Algorithm 1. Compared with the original version of FreeAT, we replace the classification loss $L$ with the detection loss $L_d$ (see Eq. (1)) and initialize the model with upstream adversarially pre-trained backbones. With FreeAT, the object detector can update the parameters per backpropagation. Thus, the cost of AT can be reduced to be similar to that of standard training.

---

**Algorithm 1** "Free" Adversarial Training on object detection

---

**Require:** Dataset $\mathcal{D}$, perturbation intensity $\epsilon$, replay parameter $m$, model parameters $\theta$, epoch $N_{\mathrm{ep}}$
 1: Initialize $\theta$ with upstream adversarial pre-training
 2: $\delta \leftarrow \mathbf{0}$
 3: **for** epoch $= 1, \ldots, N_{\mathrm{ep}}/m$ **do**
 4:      **for** minibatch $B \sim \mathcal{D}$ **do**
 5:          **for** i $= 1, \ldots, m$ **do**
 6:              Compute gradient of loss with respect to $\mathbf{x}$
 7:                  $\mathbf{g}_{\mathrm{adv}} \leftarrow \mathbb{E}_{\mathbf{x} \in B}[\nabla_{\mathbf{x}} L_d(\mathbf{x} + \delta, \theta)]$
 8:              Update $\theta$ with an optimizer
 9:                  $\mathbf{g}_{\theta} \leftarrow \mathbb{E}_{\mathbf{x} \in B}[\nabla_{\theta} L_d(\mathbf{x} + \delta, \theta)]$
10:                  update $\theta$ with $\mathbf{g}_{\theta}$ and the optimizer
11:              Use $\mathbf{g}_{\mathrm{adv}}$ to update $\delta$
12:                  $\delta \leftarrow \delta + \epsilon \cdot \mathrm{sign}(\mathbf{g}_{\mathrm{adv}})$
13:                  $\delta \leftarrow \mathrm{clip}(\delta, -\epsilon, \epsilon)$
14:          **end for**
15:      **end for**
16: **end for**

---

# D  OTHER RESULTS ON PASCAL VOC

## D.1  OTHER VISUALIZATION COMPARISON RESULTS

Three other visualization comparisons of the detection results are shown in Fig. A5 at the end of the Appendix as Fig. A5 is relatively large. Similar to Fig. 1, the detector with our recipe performed significantly better on the three images than that with the recipe of previous works (Zhang & Wang, 2019; Chen et al., 2021; Dong et al., 2022).

## D.2  COMPARISON RESULTS BETWEEN FREEAT AND PGD-AT

We compared the results of detectors trained with FreeAT and the full PGD-AT (Madry et al., 2018). The full PGD-AT used PGD with iterative steps $t = 10$ and step size $\alpha = 2$, which required $20\times$ equivalent training time for $2\times$ training schedule. The results shown in Table A8 indicate that FreeAT with $m = 4$ achieved comparable detection accuracy with the full PGD-AT under various attacks. In addition, we performed an ablation study on the replay parameter $m$. Table A8 shows that FreeAT with $m = 4$ achieved the best detection accuracy under attacks.

Table A8: The evaluation results of Faster R-CNN trained with different AT settings on PASCAL VOC.

| Training Method | Benign | $A_{\text{cls}}$ | $A_{\text{reg}}$ | $A_{\text{cwa}}$ |
|---|---|---|---|---|
| FreeAT($m = 2$) | **75.7** | 25.7 | 45.9 | 26.7 |
| FreeAT($m = 4$) | 69.7 | 32.2 | **51.8** | 34.4 |
| FreeAT($m = 6$) | 64.7 | 31.1 | 49.7 | 33.8 |
| PGD-AT($t = 10$) | 68.9 | **32.4** | 51.3 | **34.6** |

# E  OTHER DETAILS AND RESULTS ON MS-COCO

## E.1  IMPLEMENTATION DETAILS

Unless otherwise specified, the upstream adversarially pre-trained backbones were taken from Salman et al. (2020) (for ResNet-50) and Liu et al. (2023) (for ConvNeXt-T). Other training settings basically followed the default setting in MMDetection. All experiments were conducted on 8 NVIDIA 3090 GPUs with a batch size of 16. The detectors were optimized by AdamW with an initial learning rate of $1 \times 10^{-4}$ and a weight decay of 0.1. For the learning rate schedule, the detectors used multi-step decay that scaled the learning rate by 0.1 after the 20th epoch. The input images were resized to have their shorter side being 800 and their longer side less or equal to 1333.

## E.2  FULL RESULTS UNDER $A_{\text{reg}}$ AND $A_{\text{cwa}}$

The full evaluation results (under $A_{\text{reg}}$ and $A_{\text{cwa}}$) of different object detectors for Tables 5 and 3 are shown in Tables A9 and A10, respectively.

## E.3  ERROR ANALYSIS OF DIFFERENT DETECTORS

We further analyze the errors caused by the attacks by comparing the error distribution of these detectors before and after the attack in Fig. A4. The error distribution was evaluated by the COCO

Table A9: The evaluation results of object detectors with two backbones ResNet-50 (R-50) and ConvNeXt-T (X-T) on MS-COCO. Detectors were trained by VANAT with our recipe.

| Detector | Backbone | $A_{\text{reg}}$ | | | | | | $A_{\text{cwa}}$ | | | | | |
|---|---|---|---|---|---|---|---|---|---|---|---|---|---|
| | | AP | AP$_{50}$ | AP$_{75}$ | AP$_S$ | AP$_M$ | AP$_L$ | AP | AP$_{50}$ | AP$_{75}$ | AP$_S$ | AP$_M$ | AP$_L$ |
| Faster R-CNN | R-50 | 19.7 | 40.5 | 17.0 | 9.7 | 21.3 | 27.7 | 15.1 | 29.3 | 14.0 | 6.2 | 15.7 | 22.4 |
| | X-T | 23.3 | **46.4** | 20.8 | 12.6 | 24.7 | 33.2 | 19.0 | **35.9** | 18.1 | 8.1 | 19.5 | 28.2 |
| FCOS | R-50 | 27.1 | 44.4 | 28.0 | 14.0 | 29.9 | 36.4 | 14.7 | 24.0 | 15.1 | 6.0 | 15.4 | 21.5 |
| | X-T | 31.4 | **50.8** | 32.2 | 17.4 | 34.1 | 43.1 | 18.9 | **30.4** | 19.5 | 7.7 | 19.4 | 28.1 |
| DN-DETR | R-50 | 25.0 | 43.8 | 25.0 | 8.2 | 25.7 | 41.9 | 15.9 | 27.4 | 15.8 | 4.9 | 16.6 | 25.9 |
| | X-T | 27.9 | **47.4** | 28.2 | 9.2 | 28.8 | 47.1 | 18.7 | **30.9** | 18.7 | 5.9 | 19.6 | 31.1 |

Table A10: The evaluation results of object detectors under VANAT (two different training recipes, Beni-AT and Our-AT) and standard training (STD) on MS-COCO. The results of $AP_{50}$ are shaded as it is a more practical metric.

| Detector | Method | $A_{\mathrm{reg}}$ | | | | | | $A_{\mathrm{cwa}}$ | | | | | |
|---|---|---|---|---|---|---|---|---|---|---|---|---|---|
| | | AP | $AP_{50}$ | $AP_{75}$ | $AP_S$ | $AP_M$ | $AP_L$ | AP | $AP_{50}$ | $AP_{75}$ | $AP_S$ | $AP_M$ | $AP_L$ |
| Faster R-CNN | STD | 0.0 | 0.1 | 0.0 | 0.0 | 0.0 | 0.0 | 0.0 | 0.0 | 0.0 | 0.0 | 0.0 | 0.1 |
| | Beni-AT | 15.7 | 33.7 | 12.7 | 8.6 | 16.9 | 21.1 | 11.1 | 22.1 | 10.0 | 4.6 | 11.6 | 16.0 |
| | Our-AT | 19.7 | **40.5** | 17.0 | 9.7 | 21.3 | 27.7 | 15.1 | **29.3** | 14.0 | 6.2 | 15.7 | 22.4 |
| FCOS | STD | 1.8 | 4.8 | 1.2 | 0.0 | 0.5 | 4.0 | 0.5 | 1.4 | 0.3 | 0.2 | 0.7 | 1.1 |
| | Beni-AT | 20.2 | 33.9 | 20.6 | 10.6 | 22.1 | 26.7 | 10.1 | 16.6 | 10.2 | 4.5 | 10.6 | 14.5 |
| | Our-AT | 27.1 | **44.4** | 28.0 | 14.0 | 29.9 | 36.4 | 14.7 | **24.0** | 15.1 | 6.0 | 15.4 | 21.5 |
| DN-DETR | STD | 2.4 | 6.4 | 1.5 | 0.3 | 2.4 | 5.1 | 0.2 | 0.5 | 0.2 | 0.1 | 0.2 | 0.6 |
| | Beni-AT | 23.5 | 43.6 | 22.7 | 7.4 | 22.3 | 39.3 | 10.0 | 17.5 | 9.5 | 3.4 | 10.6 | 16.0 |
| | Our-AT | 25.0 | **43.8** | 25.0 | 8.2 | 25.7 | 41.9 | 15.9 | **27.4** | 15.8 | 4.9 | 16.6 | 25.9 |

analysis tool[3]. We found that for all three detectors, *the attacks mainly caused false negative (FN) errors and background errors (BG) of detectors.* This conclusion is consistent with the visualization, *e.g.*, the attack caused the detector to confuse the background as objects (*i.e.*, BG) in Fig. 1.

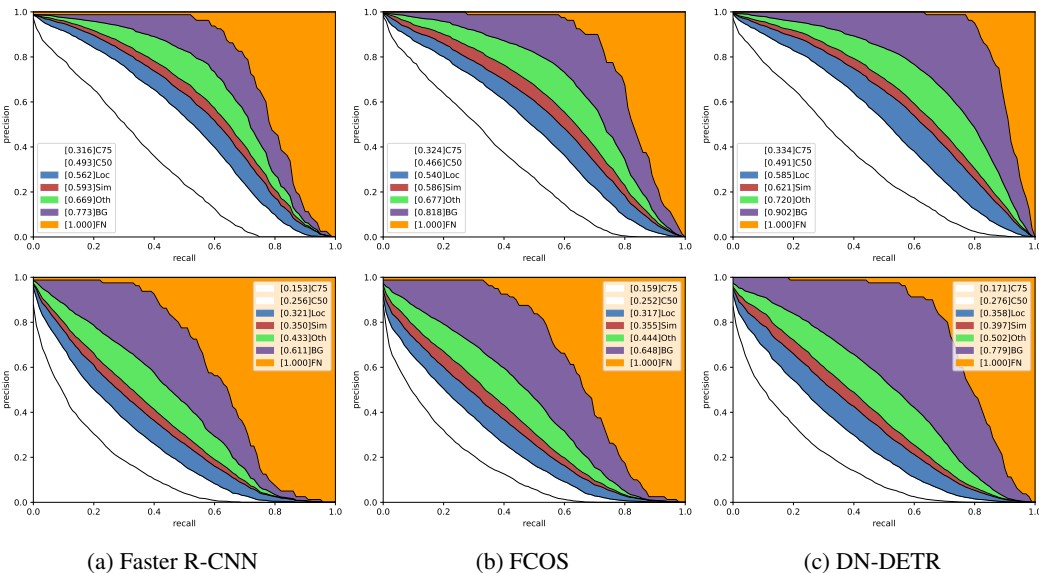

(a) Faster R-CNN  (b) FCOS  (c) DN-DETR

Figure A4: Breakdown of errors on benign examples (upper) and $A_{\mathrm{cls}}$ adversarial examples (lower). Each curve is obtained by gradually relaxing the evaluation criteria. The severity of a particular error is reflected by the area between the curves, which is indicated in the legend. The errors are categorized as follows: C75: PR curve at IoU of 0.75, corresponding to $AP_{50}$. C50: PR curve at IoU of 0.75, corresponding to $AP_{75}$. Loc: false positives (FP) caused by poor localization. Sim: FP caused by confusion with similar objects. Oth: FP caused by confusion with other objects. BG: FP caused by confusion with background or unlabeled objects. FN: false negatives.

## E.4 DIFFERENT UPSTREAM ADVERSARIAL PRE-TRAINING METHODS

We investigated the influence of different upstream adversarial pre-training manners for the same backbone network. Both Debenedetti et al. (2022) and Liu et al. (2023) adversarially trained the same ConvNeXt-T network but with different AT recipes. They achieved 44.4% and 48.8% accuracy on ImageNet under AA, respectively. We used their checkpoints to initialize the backbone of different detectors and then performed VANAT with our recipe. The results are shown in Table A11. We found

---

[3]http://cocodataset.org/#detection-eval

that a better upstream adversarial pre-training recipe led to better detection performance. Thus, we urge the community to explore the potential of large-scale pre-training in adversarial robustness more.

Table A11: The evaluation results of object detectors with the backbone (ConvNeXt-T) pre-trained with different AT manners. Detectors were trained on COCO by VANAT with our recipe.

| Detector | Pre-training method | Benign | | | | | $A_{cls}$ | | | | | $A_{reg}$ | $A_{cwa}$ |
|---|---|---|---|---|---|---|---|---|---|---|---|---|---|
| | | AP | AP$_{50}$ | AP$_S$ | AP$_M$ | AP$_L$ | AP | AP$_{50}$ | AP$_S$ | AP$_M$ | AP$_L$ | AP$_{50}$ | AP$_{50}$ |
| Faster R-CNN | Debenedetti et al. (2022) | 32.6 | 52.9 | 17.1 | 34.8 | 45.4 | 17.5 | 29.8 | 6.1 | 17.1 | 27.2 | 43.1 | 33.2 |
| | Liu et al. (2023) | 34.3 | 55.4 | 19.3 | 36.9 | 46.8 | 19.0 | **32.4** | 7.4 | 19.5 | 28.7 | **46.4** | **35.9** |
| FCOS | Debenedetti et al. (2022) | 33.8 | 51.4 | 17.9 | 36.6 | 46.6 | 18.5 | 29.5 | 7.2 | 18.3 | 27.9 | 48.4 | 28.2 |
| | Liu et al. (2023) | 35.6 | 53.8 | 20.1 | 38.2 | 48.1 | 19.8 | **31.7** | 8.6 | 20.2 | 29.0 | **50.8** | **30.4** |
| DN-DETR | Debenedetti et al. (2022) | 33.9 | 51.6 | 13.9 | 36.1 | 53.6 | 17.9 | 28.9 | 5.9 | 17.9 | 29.3 | 46.0 | 27.6 |
| | Liu et al. (2023) | 34.2 | 52.0 | 13.4 | 36.6 | 54.7 | 19.9 | **32.0** | 7.1 | 20.9 | 32.8 | **47.4** | **30.9** |

## F DETAILS ON THE REDESIGNED DETECTORS

To allocate more computation to the backbone network while maintaining the overall inference speed (FPS), we modified the depth and width (channel) of the object detector configurations. Specifically, we increased the number of layers in the backbone networks. Meanwhile, for Faster R-CNN and FCOS, the number of channels of the detection head were reduced, and for DN-DETR, the number of layers of the detection head was reduced. We made the following modifications to the default configurations:

- Backbone: We used ConvNeXt-T as the backbone of the three detectors in our experiments and modified the number of blocks in each stage from (3, 3, 9, 3) to (3, 3, 12, 3). The upstream adversarial pre-training for the modified ConvNeXt-T used the same training setting as that of Liu et al. (2023).

- Faster R-CNN head: We reduced the number of channels in the RPN and RoI head from 256 to 192.

- FCOS head: We reduced the number of channels in the FCOS head from 256 to 192.

- DN-DETR head: We reduced the number of Transformer layers of the Transformer encoder from 6 to 3.

As shown in Table A12, by comparison with the default detector configurations (note that the default object detector configurations in MMdetection have been highly optimized), we surprisingly found that these modifications significantly improved the detection accuracy of *all detectors* on benign examples and *all types* of adversarial samples. Furthermore, as presented in Table A13, our modifications also boosted the actual inference speed (FPS) of the detectors to varying degrees. We also report the theoretical FLOPs and the number of parameters in Table A13, where our method likewise presents an overall advantage. Note that these modifications are intended to validate the usefulness of our conclusion and could be further improved, which is beyond the scope of this work.

Table A12: Detailed comparison of detection accuracy on benign and adversarial examples. Symbol * denotes our designed detectors with new computation allocation.

| Detector | Benign | | | | | | $A_{cls}$ | | | | | | $A_{reg}$ | $A_{cwa}$ |
|---|---|---|---|---|---|---|---|---|---|---|---|---|---|---|
| | AP | AP$_{50}$ | AP$_{75}$ | AP$_S$ | AP$_M$ | AP$_L$ | AP | AP$_{50}$ | AP$_{75}$ | AP$_S$ | AP$_M$ | AP$_L$ | AP$_{50}$ | AP$_{50}$ |
| Faster R-CNN | 34.3 | 55.4 | 36.6 | 19.3 | 36.9 | 46.8 | 19.0 | 32.4 | 19.3 | 7.4 | 19.5 | 28.7 | 46.4 | 35.9 |
| Faster R-CNN* | 35.1 | **56.5** | 37.4 | 19.6 | 37.9 | 47.5 | 19.7 | **33.3** | 20.2 | 7.8 | 20.0 | 30.0 | **47.3** | **37.1** |
| FCOS | 35.6 | 53.8 | 37.7 | 20.1 | 38.2 | 48.1 | 19.8 | 31.7 | 20.5 | 8.6 | 20.2 | 29.0 | 50.8 | 30.4 |
| FCOS* | 36.6 | **55.0** | 39.0 | 21.2 | 39.9 | 49.0 | 21.0 | **33.3** | 21.7 | 9.0 | 21.8 | 30.7 | **52.2** | **31.9** |
| DN-DETR | 34.2 | 52.0 | 36.1 | 13.4 | 36.6 | 54.7 | 19.9 | 32.0 | 20.3 | 7.1 | 20.9 | 32.8 | 47.4 | 30.9 |
| DN-DETR* | 34.7 | **53.0** | 36.8 | 14.4 | 37.8 | 54.4 | 20.3 | **32.8** | 20.6 | 6.9 | 21.4 | 32.7 | **47.8** | **31.7** |

Table A13: Detailed comparison of parameters and computational cost. Symbol $*$ denotes our designed detectors with new computation allocation. FPS was tested on an NVIDIA 3090 GPU. Note that DETR-like models usually have smaller theoretical FLOPs than other detectors, which was also observed in previous work (Meng et al., 2021; Li et al., 2022b).

| Detector | Backbone | | Head | | Sum | | FPS |
|---|---|---|---|---|---|---|---|
| | #Param. (M) | FLOPs (G) | #Param. (M) | FLOPs (G) | #Param. (M) | FLOPs (G) | |
| Faster R-CNN | 27.6 | 91.0 | 17.7 | 118.1 | 45.3 | 209.1 | 25.4 |
| Faster R-CNN$^*$ | 31.2 | 105.4 | 7.3 | 64.3 | 38.5 | 169.7 | **25.6** |
| FCOS | 27.6 | 91.0 | 8.2 | 119.0 | 35.8 | 210.0 | 24.5 |
| FCOS$^*$ | 31.2 | 105.4 | 4.8 | 68.0 | 36.0 | 173.4 | **25.3** |
| DN-DETR | 27.6 | 91.0 | 20.2 | 12.4 | 47.8 | 103.4 | 20.0 |
| DN-DETR$^*$ | 31.2 | 105.4 | 16.0 | 8.8 | 47.2 | 114.2 | **20.1** |

## G  THE ATTACK SETTING OF PANOPTIC SEGMENTATION

Following the common attack setting on ImageNet, $\epsilon = 4$ was used for panoptic segmentation. Like those introduced in Sec. 3, we found previous work on panoptic segmentation (Daza et al., 2022) also used a weak attack so that the adversarial robustness they reported could be overestimated. However, as the code and the adversarially trained checkpoint were not released, we cannot perform our reliable attack evaluation on their method directly. Instead, we compared our attack with their attack on the same standardly trained models (STD). The results are shown in Table A14. We found that our attack reduced the Panoptic Quality (PQ) of STD to 1.5% while their attack only reduced PQ to 12.3%, indicating that the attack we used for evaluation was a reliable and strong attack compared with Daza et al. (2022).

Table A14: Results of standardly trained panoptic segmentation models (STD) under different attacks. The results of Daza et al. (2022) are copied from their original paper.

| **Detector** (STD) | **Attack method** | PQ | SQ | RQ |
|---|---|---|---|---|
| PanopticFPN | Daza et al. (2022) | 12.3 | 64.0 | 14.6 |
| | $A_{\text{cls}}$ | 1.5 | 48.4 | 2.4 |

## H  A FURTHER DISCUSSION ON ROBUST OBJECT DETECTORS

As described in Sec. 6, we have designed several adversarially robust object detectors based on our findings. Take the following as examples, we discuss how the adversarially robust object detectors may be further improved in the future based on our study.

- Firstly, our work encourages the community to explore the potential of large-scale pre-training on adversarial robustness more, which has shown great success in improving benign accuracy of downstream tasks (He et al., 2020; 2022). We note that most of the current published works in the adversarial training area still stay at the CIFAR-10 (Krizhevsky et al., 2009) level and large-scale adversarial pre-training is relatively under-explored.

- Secondly, our other findings about the main errors caused by the attack (*e.g.*, small object, FN, and BG errors) could encourage future works to focus on designing new techniques, *e.g.*, small-object-specific AT and advanced foreground-background-friendly modules to improve these weaknesses of object detectors.

- Thirdly, our finding about transfer attacks on object detectors (transferring between detection-specific modules is easier than transferring between backbone networks) may inspire better model ensemble attack and defense on object detectors. We note that previous studies such as Hu et al. (2022) mainly performed ensemble on different detection-specific modules instead of different backbones.

Overall, we believe this study together with these findings serves as a basis for building better adversarially robust object detectors in the future.

# I  ETHICAL INFLUENCE

Our method increases the robustness of object detectors so that they are potentially less attackable when they are used for malicious purposes. For example, this could lead to more effective surveillance, potentially encroaching upon personal privacy when abused. But in general, we believe the concrete positive impact on safety outweighs the potential negative impacts.

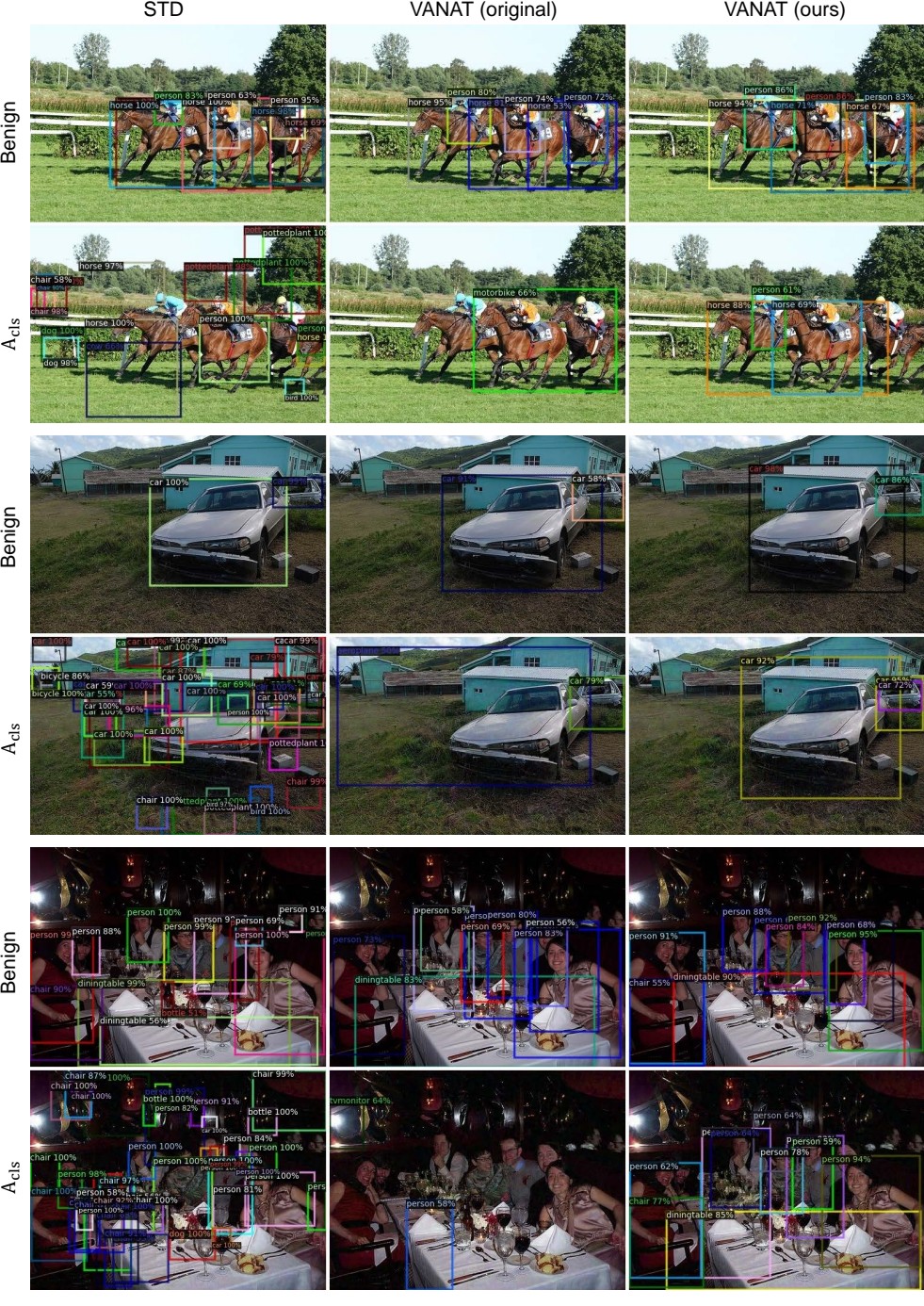

Figure A5: Three other visualization comparisons of the detection results on benign images (upper) and on $A_{\mathrm{cls}}$ adversarial images (lower), with three training methods STD (left), VANAT with the recipe of previous work (medium), and VANAT with our recipe (right). Faster R-CNN was used here.

