# OpenReview forum: "On the Importance of Backbone to the Adversarial Robustness of Object Detectors"
_ICLR.cc/2024/Conference — ICLR 2024 Conference Withdrawn Submission_

### Official Review · Reviewer_fyi7 · 2023-10-29

**Soundness:** 3 good
**Presentation:** 3 good
**Contribution:** 3 good
**Rating:** 5
**Confidence:** 3

**Summary:**

In this paper, the authors find that existing works on improving the adversarial robustness of object detectors give a false sense of security. They propose a simple yet effective recipe for fast adversarial fine-tuning on object detectors with adversarially pre-trained backbones. They explore the potential of different modern object detectors to improve adversarial robustness using our recipe and demonstrated interesting findings. The empirical results set a new milestone for adversarially robust object detection.

**Strengths:**

- Propose a simple yet effective recipe for fast adversarial fine-tuning on object detectors with adversarially pre-trained backbones
- Conduct large-scale empirical study

**Weaknesses:**

- Lack of practicality

In this paper, the authors are motivated by some real-world practical security-sensitive applications such as autonomous driving and video surveillance. However, later on, there is no such evaluation. Thus, it lacks a thorough discussion on the transferability of their findings across different applications. For instance, the applicability of their observations to tasks such as real-time detection in autonomous vehicles unclear. Future iterations of the work could benefit from a section discussing the limitations and potential practicality of the findings, supplemented by experiments in diverse application scenarios.

- Lack of overhead evaluation

The authors propose modifications to the detector architectures (i.e., the backbone networks) to improve adversarial robustness. However, the discussion on the computational overheads, particularly in real-time applications, remains unclear. A more detailed analysis of the trade-offs between increased robustness and computational efficiency will strengthen the paper practical contributions since this paper is motivated by the practical real-world applications such as autonomous driving.

- The methodology and setup of the re-evaluation is not convincing

The method and setup for the re-evaluation are not comprehensive. For instance, their evaluation may be very specific to the dataset, which means that the findings are not general. Also, it is unclear whether their selected re-evaluated targets are representative or state-of-the-art. Without such justification, it is unclear whether their findings are representative or not. Thus, I would recommend the authors to provide more discussion.

**Questions:**

Provide more evaluation and discussion on the practicality, overhead, and justification for evaluation methodology and setup.

---

> ### Author Response · Authors · 2023-11-19
> **Response to Reviewer fyi7 (1/2)**
>
> We appreciate that the reviewer thinks our method is simple yet effective, together with a large-scale empirical study. Our responses to  the weaknesses and your questions are below:
>
> **Q1: Lack of practicality. In this paper, the authors are motivated by some real-world practical security-sensitive applications such as autonomous driving and video surveillance. However, later on, there is no such evaluation. Thus, it lacks a thorough discussion on the transferability of their findings across different applications. For instance, the applicability of their observations to tasks such as real-time detection in autonomous vehicles is unclear. Future iterations of the work could benefit from a section discussing the limitations and potential practicality of the findings, supplemented by experiments in diverse application scenarios.**
>
> We appreciate the reviewer's comments regarding the practicality of our work. We would like to clarify that although autonomous driving and video surveillance are indeed specific safety-critical applications of the object detection task, our primary focus in this work is on the object detection task itself. Since the two applications themselves involve additional and complex modules, a detailed evaluation of them is beyond the scope of this study. We clarify that we mentioned these applications in the manuscript just to illustrate the importance of investigating the adversarial robustness of object detectors.
>
> Besides practical considerations, the object detection task itself is one of the most fundamental and important tasks in computer vision, and the complexity of object detectors, as evident in Table 4, calls for a systematic study to enhance their robustness. We note that previous works have not sufficiently explored their robustness and give a false sense of security on object detectors (as shown in Section 3), which also motivates our comprehensive investigation.
>
>
>
> **Q2: Lack of overhead evaluation. The authors propose modifications to the detector architectures (i.e., the backbone networks) to improve adversarial robustness. However, the discussion on the computational overheads, particularly in real-time applications, remains unclear. A more detailed analysis of the trade-offs between increased robustness and computational efficiency will strengthen the paper practical contributions since this paper is motivated by practical real-world applications such as autonomous driving.**
>
> Since we have proposed in our paper two modifications on the backbone network to improve the adversarial robustness, for clarity, we explain them in turn:
>
> - Use an adversarially pre-trained backbone instead of a benignly pre-trained backbone (Section 4). In this case, since our modification only involves changing the pre-trained weights of the backbone network, without altering its architecture, it **does not increase any overhead** of the inference.
> - Allocate more computation to the backbone network and reduce the computation of detection-specific modules (Section 6). This modification was to show the application of our finding that backbone networks play a more important role than detection-specific modules. Table 6 in Appendix F shows the evaluation of both the efficiency (FPS) and robustness (AP). Importantly, the computational reallocation **maintains and even improves computational efficiency**. This is because our method optimizes the allocation of computational resources: *allocating more computation to the backbone and reducing the computation of detection-specific modules inspired by our findings so that overall inference speed is not sacrificed (Section 6).
>
> **In summary, neither of our proposed modifications increases the inference overhead.** We will emphasize this more explicitly in the revised version.

---

> ### Author Response · Authors · 2023-11-19
> **Response to Reviewer fyi7 (2/2)**
>
> **Q3: The methodology and setup of the re-evaluation is not convincing. The method and setup for the re-evaluation are not comprehensive. For instance, their evaluation may be very specific to the dataset, which means that the findings are not general. Also, it is unclear whether their selected re-evaluated targets are representative or state-of-the-art. Without such justification, it is unclear whether their findings are representative or not. Thus, I would recommend the authors to provide more discussion.**
>
> Thanks for raising this concern. We would like to clarify that our evaluation is indeed comprehensive. In fact, our evaluation includes two aspects:
>
> - First, we re-evaluate all previous methods on adversarially robust detectors in Section 3. These methods represented the SOTA methods in this area, to the best of our knowledge. As these SOTA methods performed their main experiments on PASCAL VOC, our re-evaluation was also performed on this dataset. To demonstrate the consistency of the conclusion from our re-evaluation, here we performed an additional similar re-evaluation on the MS-COCO dataset using Faster R-CNN (for AARD we used SSD of their released checkpoint). The results are shown below (for conciseness, we only show $AP_{50}$). The conclusion is similar to those on PASCAL VOC (Section 3.2): CWAT and MTD did not show significant improvements over VANAT, and AARD was fully broken with the stronger attack.
>
> - Second, we evaluated the potential of several different modern object detectors in improving adversarial robustness in Section 5. In this evaluation, we focused on the MS-COCO dataset due to computational constraints. However, it is important to note that MS-COCO is a large-scale dataset that includes almost all the categories from the PASCAL VOC dataset. Therefore, the findings from the MS-COCO evaluation are applicable to the PASCAL VOC dataset as well. Additionally, we highlight that modern detectors, such as FCOS and DN-DETR, also only reported the results on the MS-COCO dataset in their papers. Hence, we believe that the evaluation and findings on MS-COCO are comprehensive enough to provide valuable conclusions for the broader object detection community.
>
> ||||||
> |-|-|-|-|-|
> |Method|Benign|$A_{cls}$|$A_{reg}$|$A_{cwa}$|
> |CWA|46.3|24.2|38.6|26.8|
> |MTD|49.4|24.8|40.5|29.3|
> |VANAT|**49.3**|**25.5**|**40.5**|**29.3**|
> |AARD(SSD)|36.7|0.0|0.0|0.0|
>
> **We are happy to answer any further questions related to our work.**

---

### Official Review · Reviewer_wtFb · 2023-10-31

**Soundness:** 3 good
**Presentation:** 4 excellent
**Contribution:** 2 fair
**Rating:** 5
**Confidence:** 5

**Summary:**

In this paper, the problem of adversarial robustness for object detection task is investigated and the study has some practical value. The authors improve the robustness of the backbone network through adversarial training methods, and the profile improves the stability of the prediction results on the detection task. The authors validate this idea through a large number of experiments.

**Strengths:**

1. Extensive experimental validation. The article conducts experiments on several publicly available datasets as well as detectors to comprehensively validate the effectiveness and robustness of the proposed method.
2. Clear diagrams. The figures and tables are well-designed and help readers understand the content of the article.
3. Clear background introduction. The article provides a thorough review of related work on object detection defense in the introduction section, which provides readers with good background knowledge.

**Weaknesses:**

1. Overall lack of innovation. Although the authors reveal the importance of the backbone network in the overall robustness of the model, this idea is not uncommon [1]. Similar ideas have been proposed by related scholars to illustrate the importance of backbone networks. Since this is a backbone network analysis related to object detectors, the authors should provide more backbone network analyses related to object detectors, e.g., EfficientNet [2], MobileNet [3], DarkNet [4]. In addition, the authors did not propose too novel improvements in the adversarial training phase of the backbone network.
2. more validation for different tasks. Since the backbone network has a huge impact on the detector, I think the authors' approach should not be limited to the detector task, but the authors can try the results of image segmentation, as well as image categorization to evaluate the enhancement of the robust backbone network. I believe that the idea that robust backbone networks improve the robustness of the detector is applicable on a wide range of tasks. Since as an universal idea, I think the authors should not limit themselves to one task.
3. How do the authors consider about the gap between pre-trained models and downstream task features? Why are robust features trained using existing adversarial training methods a good feature for detectors? I look forward to the authors' reply about the inspiration and thoughts on the object detection task for robust feature design for backbone networks.

[1] "Do Adversarially Robust ImageNet Models Transfer Better?" NeurIPS Proceedings. 2020.
[2] "Efficientnet: Rethinking model scaling for convolutional neural networks." arXiv preprint arXiv:1905.11946.
[3] "Mobilenets: Efficient convolutional neural networks for mobile vision applications." arXiv preprint arXiv:1704.04861.
[4] "You only look once: Unified, real-time object detection." In Proceedings of the IEEE conference on computer vision and pattern recognition, pp. 779-788.

**Questions:**

Please refer to the weaknesses section.

---

> ### Author Response · Authors · 2023-11-19
> **Response to Reviewer wtFb (1/2)**
>
> We would first like to thank the reviewer for the time spent on our work and the valuable comments. Below are our responses to the concerns:
>
>
>
> **Q1: Overall lack of innovation. Although the authors reveal the importance of the backbone network in the overall robustness of the model, this idea is not uncommon [1]. Similar ideas have been proposed by related scholars to illustrate the importance of backbone networks. Since this is a backbone network analysis related to object detectors, the authors should provide more backbone network analyses related to object detectors, e.g., EfficientNet, MobileNet, DarkNet. In addition, the authors did not propose too novel improvements in the adversarial training phase of the backbone network.**
>
> First, we would like to clarify that [1] completely differs from our work **in the research goal**. The main contribution of [1] lies in finding that adversarially robust ImageNet classifiers yield better accuracy on **clean examples** of other classification datasets in transfer learning settings. **However, they did not report any results or show any claims of whether adversarially robust ImageNet classifiers can boost the adversarial robustness of downstream dense-prediction tasks.** In contrast, our work focuses on the **adversarial robustness** of object detectors under attacks. To the best of our knowledge, the vitalness of the adversarial robustness of backbone networks to the adversarial robustness of downstream tasks has been neglected for a long time and has never been demonstrated before our work.
>
> Second, we want to clarify that the primary focus of this work is to investigate the factors that affect the robustness of a detector, rather than providing a benchmark comparison of different backbone architectures for robustness. Besides, the backbones you mentioned have not been validated for robustness on large-scale upstream classifications such as ImageNet-1K, and there are no available adversarially pre-trained checkpoints for those backbones, either.  Due to the fact that our method relies on upstream adversarial pre-training, we regret that we cannot provide extensive experiments on these backbones, which is also beyond the scope of this paper.
>
> Third, we note that investigating the training recipe is important in the domain of adversarial robustness for classification [2][3] but it has not been explored on downstream tasks like object detection. Such an exploration would be crucial for improving the robustness of object detectors. Considering the remarkable SOTA results we have achieved (see Table 2), we respectfully disagree that our improvements in the adversarial training phase (e.g., the utilization of AdamW, backbone learning rate scaling, etc.) are not novel, especially in the context that both [2] and [3] suggest that using SGD optimizer with momentum in AT can obtain better adversarial robustness for classifiers than other optimizers.
>
> **Q2: More validation for different tasks. Since the backbone network has a huge impact on the detector, I think the authors' approach should not be limited to the detector task, but the authors can try the results of image segmentation, as well as image categorization to evaluate the enhancement of the robust backbone network. I believe that the idea that robust backbone networks improve the robustness of the detector is applicable on a wide range of tasks.**
>
> Thanks for your suggestion. We mainly focus on object detection since it is one of the most fundamental tasks in computer vision with lots of security-critical applications. Besides, object detectors themselves are often complicated with multiple components (Table 4) and thus deserve a systematic study to improve their robustness.
>
> But we agree with you that adversarially robust backbone networks should also be effective in improving robustness for other tasks. In fact, we did conduct experiments on other tasks (*i.e.*, panoptic segmentation) to verify the generalizability of our conclusions. The details are presented in Section 6. We chose panoptic segmentation since it is a very challenging image segmentation task and requires solving both instance and semantic segmentation tasks simultaneously. We will make it clearer in the revised version.
>
> [1] "Do Adversarially Robust ImageNet Models Transfer Better?". NeurIPS, 2020.
>
> [2]: Pang T, Yang X, Dong Y, et al. Bag of Tricks for Adversarial Training. ICLR, 2020.
>
> [3]: Mo Y, Wu D, Wang Y, et al. When adversarial training meets vision transformers: Recipes from training to architecture[C]. NeurIPS, 2022.

---

> ### Author Response · Authors · 2023-11-19
> **Response to Reviewer wtFb (2/2)**
>
> **Q3: How do the authors consider about the gap between pre-trained models and downstream task features? Why are robust features trained using existing adversarial training methods a good feature for detectors? I look forward to the authors' reply about the inspiration and thoughts on the object detection task for robust feature design for backbone networks.**
>
> We think these are good questions, and our insight about them includes the following two aspects:
>
> - The benefits of adversarial pre-training could outweigh the potential conflicts. In the object detection field, for clean images (i.e., without adversarial noise), pre-training on the upstream classification tasks and then fine-tuning on object detection tasks is the de facto practice [4][5][6]. Thus the inconsistency problem also exists for almost all standard-trained object detectors (i.e. without AT). However, in practice, the benefits of pre-training still outweigh the potential conflicts. For adversarial examples, we think the case could be similar.  Thanks to the large-scale upstream pre-training data, adversarial pre-training could enable the network to learn some general robust features, which is also beneficial to the robustness of downstream tasks.
> - On the other hand, training on downstream tasks with small-scale datasets may cause the backbone network to lose these pre-trained robust features.  To ensure that the original adversarial robustness of backbones (robust feature) is preserved during downstream fine-tuning and the detection-specific modules can be trained in the usual way, we decay the learning rate of the backbone by a factor of 0.1 when performing AT with object detectors on the downstream dataset. Our experimental results in Table 2 have shown that this method is indeed effective, and we think further exploration could be a promising direction. In addition, allocating more computation to the backbone while reducing that of detection-specific modules (Section 6 and Appendix F) might also be helpful to preserve the pre-trained robust features.
>
> Overall, we believe that pre-trained robust features (in backbones) are beneficial for enhancing the robustness of downstream tasks, and certain techniques such as the backbone learning rate scaling, as demonstrated in our proposed recipe, are crucial to fully leverage and maintain the robustness of these features.
>
>
>
> **We request the reviewer to kindly re-evaluate the contribution of our work in light of the above clarification. We are also happy to answer any further questions related to our work.**
>
>
> [4] Shaoqing R, Kaiming H, et al. Faster R-CNN: towards real-time object detection with region proposal networks. NeurIPS, 2015.
>
> [5] Zhi T, Chunhua S, et al. FCOS: fully convolutional one-stage object detection. ICCV, 2019.
>
> [6] Feng L, Hao Z, et al. DN-DETR: accelerate DETR training by introducing query denoising. CVPR, 2022.

---

### Official Review · Reviewer_xgd4 · 2023-11-01

**Soundness:** 2 fair
**Presentation:** 2 fair
**Contribution:** 1 poor
**Rating:** 3
**Confidence:** 5

**Summary:**

This paper presents an adversarial defense method for object detection. The proposed new recipe mainly involves a robust backbone and FreeAT. Experiments demonstrate the effectiveness of the proposed method, especially the importance of a robust backbone.

**Strengths:**

1.[experimental evaluations] I'm glad to see the experimental evaluations are comprehensive. All types of detectors are covered in this paper -- two-stage ones, single-stage ones, as well as the DETR family.

2.[reveals deficiencies in previous works] Previous works leveraged a relatively weaker adversary for evaluation. So they are not providing as good robustness as advertised.

**Weaknesses:**

1.[important, references] Contradictory to what has been described in the introduction, using an adversarially robust backbone for downstream tasks is not completely uninvestigated. See F. Croce, N. D. Singh, M. Hein (2023). Robust Semantic Segmentation: Strong Adversarial Attacks and Fast Training of Robust Models. ICML Workshop New Frontiers in Adversarial Machine Learning. Adversarial backbone makes the adversarial training faster in order to reach a good level of robustness. And this paper tells us some similar conclusions about the importance of the backbone in the segmentation task. Based on this, it is not appropriate the finding on a robust backbone is a novelty of this paper.

2.[clarity] Without a change in architecture, how can the detector achieve a faster inference speed?

3.[novelty] While the previous adversarial training does not use adversarially robust backbones,  switching the backbone is not something new, and the robust backbone is not a contribution of this paper. Plus, a robust backbone has demonstrated its importance in previous works. Even if the previous work is a workshop paper, it is at least published instead of a preprint. So we have to take it into account. I would treat this point differently if the mentioned paper is merely an unpublished preprint -- but it is not.

**Questions:**

See weaknesses.

I'd recommend Weak reject. This is a good paper, and shows good findings. But, the proposed "new recipe" only involves two parts (1) robust backbone, and (2) fast adversarial training. The two parts are not original, and are borrowed from the classification domain. Overall, the technical contribution does not sound sufficient for ICLR, especially given that the core conclusion has been revealed by other publications already. If this paper is submitted before the arxiv timestamp of the segmentation paper mentioned above, I would have written something different. I appreciate more on original ideas instead of simply borrowing existing works from a different domain.

---

> ### Author Response · Authors · 2023-11-11
> **Response to Reviewer xgd4 (1/2)**
>
> We really appreciate that you think this is a good paper and shows good findings, with comprehensive experimental evaluations. The biggest concern is that the idea of our paper seems to borrow ideas from [1]. We will clarify this first and then reply to your other concerns.
>
>
>
> Q1. [important, references] Contradictory to what has been described in the introduction, using an adversarially robust backbone for downstream tasks is not completely uninvestigated. See F. Croce, N. D. Singh, M. Hein (2023). Robust Semantic Segmentation: Strong Adversarial Attacks and Fast Training of Robust Models. ICML Workshop. Adversarial backbone makes the adversarial training faster in order to reach a good level of robustness. And this paper tells us some similar conclusions about the importance of the backbone in the segmentation task. Based on this, it is not appropriate the finding on a robust backbone is a novelty of this paper.
>
> **We want to clarify that our idea is original and our paper has been completed and submitted to arXiv before [1]. Specifically, our paper was submitted to arXiv in May 2023 while F. Croce et al. [1] was submitted in June 2023. Therefore, It does not make sense that we borrowed ideas from [1]. Unfortunately, due to anonymity requirements,  we are unable to provide the arXiv link. To verify this fact while maintaining our anonymity, one possible solution could be to search for the title of our paper through a third party (e.g. your colleagues or friends).**
>
> In addition, we argue that [1] should be considered as a contemporaneous work of our paper, according to ICLR 2024 reviewer guidelines ([https://iclr.cc/Conferences/2024/ReviewerGuide](https://iclr.cc/Conferences/2024/ReviewerGuide)): "*the papers are considered contemporaneous if they are published (available in online proceedings) within the last four months. That means, since our full paper deadline is September 28, if a paper was published (i.e., at a peer-reviewed venue) on or after May 28, 2023, authors are not required to compare their own work to that paper. Authors are encouraged to cite and discuss all relevant papers, but they may be excused for not knowing about papers not published in peer-reviewed conference proceedings or journals···*".
>
> However, we thank you for pointing out this and are thrilled to learn about this work. Here we discuss the difference between our work and [1].
>
> - First, F. Croce et al. [1] only treated the adversarially pre-trained backbone as a tool to accelerate the convergence of AT on semantic segmentation. However, our finding is more than that. In our experiment, without upstream adversarial pre-training, directly extending the AT training schedule (e.g. 10X) resulted in a saturation of both adversarial robustness and clean accuracy (see Figure 2). This result was significantly lower than that trained for only 2X with upstream adversarial pre-training. Accordingly, we believe that the upstream adversarial pretraining is not merely a tool for accelerating AT convergence, but holds a greater potential for improving the downstream adversarial robustness.
> - Second, we revealed that *from the perspective of adversarial robustness, backbone networks play a more important role than detection-specific modules*. This finding involves not just using the adversarially robust backbone but also involves the comparison between various detection-specific modules, adversarial transferability, etc (see Section 5), which was not demonstrated by [1]. Thus our title is "On the Importance of Backbone to the Adversarial Robustness of Object Detectors" instead of "On the Importance of *Using Adversarially Robust Backbone* to the Adversarial Robustness of Object Detectors".
> - Third, our work also has other unique contributions. For example, we showed how to apply our findings in designing more robust detectors and extended our findings to panoptic segmentation tasks (Section 6). We detail these unique contributions in the response to **Q3**.
>
>
>
>
> [1] F. Croce, N. D. Singh, M. Hein. Robust Semantic Segmentation: Strong Adversarial Attacks and Fast Training of Robust Models. ICML Workshop. 2023.

---

> ### Author Response · Authors · 2023-11-19
> **Response to Reviewer xgd4 (2/2)**
>
> **Q3. While the previous adversarial training does not use adversarially robust backbones, switching the backbone is not something new, and the robust backbone is not a contribution of this paper. Plus, a robust backbone has demonstrated its importance in previous works. Even if the previous work is a workshop paper, it is at least published instead of a preprint. So we have to take it into account. I would treat this point differently if the mentioned paper is merely an unpublished preprint -- but it is not.**
>
> **Except for the contribution on the long-ignored adversarially pretrained backbones, as clarified above, our work has other comprehensive and unique contributions** compared with previous works, as listed below:
>
> - Formulating a unified reliable robustness evaluation setting for object detectors and making a thorough reevaluation of previous works, finding that previous work had given a false sense of security of object detectors (Section 3);
> - Proposing a new training recipe to better exploit the advantage of adversarial pre-trained backbones with little training cost (Section 4). We note that investigating the training recipe is important in the domain of adversarial robustness for classification [2][3] but it has not been explored on downstream tasks like detection;
> - Providing a comprehensive analysis of the robustness of object detectors and revealing several findings. This serves as a basis for building better adversarially robust object detectors (Section 5). We note that the adversarial robustness of modern detectors such as FCOS and DN-DETR has not been studied before.
> - Based on our findings, we designed several new object detectors with SOTA adversarial robustness and faster inference speed, as shown in Section 6. We also discussed how the adversarially robust object detectors may be further improved in the future based on our findings (Appendix H).
>
> **We believe our work indeed set a new milestone for adversarially robust object detection.**
>
>
>
> **Q2. Without a change in architecture, how can the detector achieve a faster inference speed?**
>
> "Without a change in architecture" refers to the experiments of the training recipe section (Section 4) and the investigation section (Section 5). "A faster inference speed'' refers to the experiments of Section 6. Inspired by our findings in Section 5, in Section 6, we design several new detectors with the principle of allocating more computation to the backbone and reducing the computation of detection-specific modules so that both the overall robustness and inference speed are improved. We will make this clearer in the revised version.
>
>
>
> **We request the reviewer to kindly re-evaluate the contribution of our work in light of the above clarification. We are also happy to answer any further questions related to our work.**
>
>
>
> [2]: Pang T, Yang X, Dong Y, et al. Bag of Tricks for Adversarial Training[C]//ICLR. 2020.
>
> [3]: Mo Y, Wu D, Wang Y, et al. When adversarial training meets vision transformers: Recipes from training to architecture[C]. NeurIPS, 2022.

---

> ### Comment · Reviewer_xgd4 · 2023-12-04
> **Review Updates**
>
> I have read all of the reviewer comments and the author's response. And I'm not convinced on the novelty level of this paper, as it is also mentioned by another reviewer.
>
> BTW, I think the authors may have misinterpreted or over-interpreted some of my comments. I meant the segmentation paper "is at least published (before the ICLR2024 submission deadline) instead of being a preprint." I just mean I can "legally" include a highly similar paper PUBLISHED several months before the ICLR2024 submission deadline in the reviewer comments, instead of expressing the intention of changing the score based on any timestamp.
>
> Workshop papers are not quite straightforward to verify on search engines. I wrote those sentences to clarify that I'm not using an UNPUBLISHED arxiv paper in the reviewer comments. Because a reviewer should not judge submissions using UNPUBLISHED preprints.
>
> Besides the mentioned segmentation paper, the other reviewer also mentioned other similar works with very similar ideas. And I think the amount of original work in this paper is still not sufficient to support its novelty.

---

### Author Response · Authors · 2023-11-19
**Response to all reviewers**

The authors are grateful to all the reviewers for the helpful comments and suggestions, and we sincerely appreciate the time and efforts the reviewers put into this manuscript, which will help us improve the quality of the revised paper. It is pretty encouraging that the reviewers think the presentation of our paper is **excellent** (Reviewer wtFb), the proposed method is **effective** and **easy to follow** (Reviewer fyi7), and the unified evaluations across various detectors are **comprehensive** (Reviewers xgd4 and wtFb), **revealing deficiencies** in previous works (Reviewer xgd4). Please find the point-to-point response to all the comments below.

---

### Author Response · Authors · 2023-11-22
**Look forward to further feedback**

**Dear reviewers**,

Thank you again for the time and effort you put into this manuscript.  With the discussion ending on Nov 22nd, we eagerly await your further feedback about our rebuttal. Have your concerns been addressed and could the rating be further adjusted based on our rebuttal? If you still have questions about our paper, we are willing to answer them and improve our manuscript.

Best, Authors